# A comprehensive examination of Nanopore native RNA sequencing for characterization of complex transcriptomes

Charlotte Soneson [1,2,5], Yao Yao[1,2], Anna Bratus-Neuenschwander[3], Andrea Patrignani[3], Mark D. Robinson [1,2] & Shobbir Hussain[4]

A platform for highly parallel direct sequencing of native RNA strands was recently described by Oxford Nanopore Technologies, but despite initial efforts it remains crucial to further investigate the technology for quantification of complex transcriptomes. Here we undertake native RNA sequencing of polyA + RNA from two human cell lines, analysing ~5.2 million aligned native RNA reads. To enable informative comparisons, we also perform relevant ONT direct cDNA- and Illumina-sequencing. We find that while native RNA sequencing does enable some of the anticipated advantages, key unexpected aspects currently hamper its performance, most notably the quite frequent inability to obtain full-length transcripts from single reads, as well as difficulties to unambiguously infer their true transcript of origin. While characterising issues that need to be addressed when investigating more complex transcriptomes, our study highlights that with some defined improvements, native RNA sequencing could be an important addition to the mammalian transcriptomics toolbox.

[1] Institute of Molecular Life Sciences, University of Zurich, 8057 Zurich, Switzerland. [2] SIB Swiss Institute of Bioinformatics, 8057 Zurich, Switzerland. [3] Functional Genomics Centre Zurich, ETHZ/University of Zurich, 8057 Zurich, Switzerland. [4] Department of Biology and Biochemistry, University of Bath, Bath BA2 7AY, UK. [5] Present address: Friedrich Miescher Institute for Biomedical Research and SIB Swiss Institute of Bioinformatics, Basel, Switzerland. Correspondence and requests for materials should be addressed to C.S. (email: charlotte.soneson@fmi.ch) or to M.D.R. (email: mark.robinson@imls.uzh.ch) or to S.H. (email: S.Hussain@bath.ac.uk)

The observed complexity of cellular mRNA splicing patterns appears to have generally expanded during the course of evolution[1], and in more advanced species, several subtly different mRNA transcript isoforms are likely to exist for most genes[2–4]. Within a biological organism, the observed pattern of mRNA splicing for a given gene also frequently varies between tissues and cell types, and can even respond to external cues or changes to the environment[5]. Thus, the ability to readily perform transcript-level functional investigations will almost certainly enrich our understanding of a number of important biological processes. To enable this to be accomplished in a reliable manner, methods that can unequivocally distinguish and quantify the presence of transcript isoforms from raw sequence reads are required.

Recently, long-read sequencing methodologies have been introduced into the transcriptomics field, offering the opportunity to directly generate individual reads that can span the full length of transcripts[6–12]. This could, for example, ameliorate problems associated with earlier technologies' needs for DNA-mediated amplification and computational transcript assembly from short sequence reads[13,14]. Notably, the long-read Oxford Nanopore Technologies (ONT) platform now also provides the ability to sequence native RNA strands directly[15]. In their study, ONT described the efficient use of native RNA sequencing to yield reliable abundance estimates of full-length transcripts from a yeast polyA + transcriptome as well as sets of standardized synthetic transcripts. However, larger transcriptome sizes, and in particular the much higher complexity of splicing patterns that can be observed in higher organisms, might pose additional challenges during such transcript-level investigations.

In this study, we apply ONT long-read native RNA sequencing to samples from two human cell lines; HAP1 and HEK293, with the primary aim of evaluating the ability to identify and quantify transcripts and genes in a complex transcriptome setting. We also perform matched ONT direct (PCR-free) cDNA sequencing as well as regular Illumina RNA-seq to enable relevant comparisons and assessments. For computational analysis, considering the lower accuracy of Nanopore sequencing, we primarily employ a reference-based approach, estimating abundances of annotated transcript isoforms and genes. An additional motivation for this is that also in situations where a reference-free approach is used for transcript identification, reference-based methods are often useful for subsequent quantification of transcript abundances. We present our findings relating to differences between the performance of a variety of analysis algorithms, and the potential advantages that current ONT direct RNA-seq brings over the traditional Illumina sequencing, as well as current limitations of the technology.

## Results

**Overall data characteristics**. We utilized three ONT library preparation workflows in this study, all having in common that RNA or cDNA molecules are sequenced directly without PCR. For our initial efforts, during which direct cDNA sequencing kits were not available from ONT, we modified the regular 2D ONT-NSK007 PCR-based workflow essentially as we previously described for NSK007 1D native genomic DNA sequencing[16,17] in order to enable 1D direct cDNA sequencing (see Methods) (Fig. 1a). We also made use of the subsequently released ONT-DCS108 kit for direct cDNA sequencing which incorporates enrichment for full-length cDNAs (Fig. 1b). Most of the data presented in this study were obtained using the ONT-RNA001 kit for native RNA sequencing (Fig. 1c).

The yield from the different ONT protocols varied between approximately 0.5 and 1.5 million unfiltered reads per library,

giving in total 1.6–4.3 million unfiltered reads per library type (Supplementary Fig. 1A). The read-length distributions were overall similar among the libraries, with a peak close to 1000 bases (Supplementary Fig. 1B), while the cDNA libraries showed higher base qualities than the native RNA libraries (Supplementary Fig. 1C). We also noticed an association between the read length and the average reported base quality, with both very short and very long reads often having lower quality (Supplementary Fig. 2). The base-level accuracy, estimated by comparing the primary genome alignments with the underlying reference sequence, showed a similar pattern as the reported average base quality (Supplementary Fig. 3).

**Genome and transcriptome alignment**. The ONT reads were aligned to the human reference genome and transcriptome using minimap2 (see Methods). The median aligned lengths for the reads in the ONT-NSK007-HAP, ONT-DCS108-HAP, ONT-RNA001-HAP, and ONT-RNA001-HEK data sets were 633, 765, 621, and 596 bases, respectively. As we aligned unfiltered reads, the alignment rates were unsurprisingly only modest, varying across protocols between 62 and 69% for the genome alignment, and from 47 to 66% for the transcriptome alignment (Fig. 2a, Supplementary Fig. 4A). As expected, the unaligned reads were enriched for low base qualities (Supplementary Fig. 5A), and thus largely represented reads that would have been classified as failed during automatic filtering. In comparison, for the Illumina libraries, STAR aligned between 89 and 94% of the reads uniquely to the genome, with an additional 2–2.5% multimapping reads. Whereas the genome alignment rates were generally only marginally higher than the transcriptome alignment rates (Fig. 2a), the ONT-DCS108-HAP libraries displayed a larger difference (64% for genome alignment vs 47% for transcriptome alignment). The reason for this disparity is unclear, but many reads aligning exclusively to the genome showed an unexpectedly high GC content (Supplementary Fig. 6).

Approximately 40% of the reads with a primary genome alignment had also at least one reported secondary genome alignment (Fig. 2b, Supplementary Fig. 4B). For most libraries, a single secondary alignment was most common, while for the ONT-DCS108-HAP libraries, a larger fraction of reads had more than five secondary genome alignments (Supplementary Fig. 7A). As expected, due to the high similarity among transcripts, the fraction of reads with reported secondary alignments increased to ~80% for the transcriptome alignment (Fig. 2b, Supplementary Fig. 4B). Again, a small number of secondary alignments was most common (Supplementary Fig. 7B). The secondary transcriptome alignment rate was only marginally affected by increasing the −p argument of minimap2 to 0.99 instead of the default 0.8 (Fig. 2b, Supplementary Fig. 4B). For a majority of the reads, the target transcripts of all primary and secondary transcriptome alignments were isoforms of the same gene (Fig. 2b, Supplementary Fig. 4B), suggesting that the main source of ambiguity is on the individual isoform level rather than on the gene level. Only a small part of the secondary alignments (typically <5% of the reads) arose due to the presence of multiple fully identical transcripts in the Ensembl reference catalog. Unavoidable secondary alignments may also be the result of reads stemming from reference transcripts that are proper subsequences of other reference transcripts. Among the 1,044,960 pairs of reference transcripts annotated to the same gene in our annotation catalog, there are 64,437 such pairs (6.2%). In these situations, a read could potentially still be considered unambiguously assignable to the shorter transcript if it is similar enough, under the assumption that all ONT reads represent full-length transcripts. Without this strong assumption, effective automated

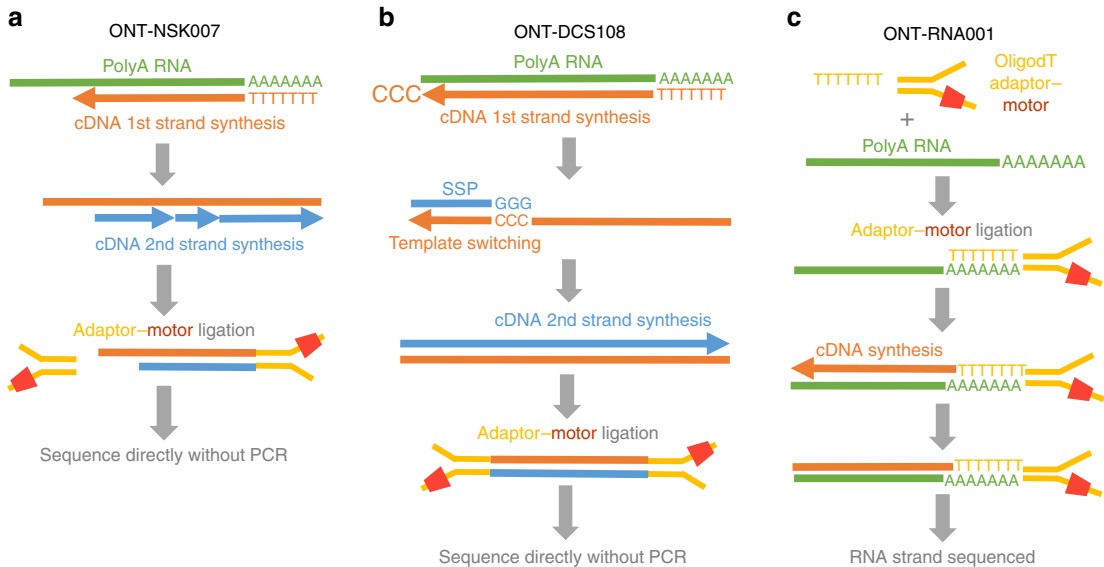

**Fig. 1** Overview of library preparation workflows used in this study. **a** In the ONT-NSK007 cDNA library preparation method, polyA RNA is used as a template for first strand cDNA synthesis which is initiated from an oligodT primer. The NEB second strand cDNA synthesis module (E6111) is then used to generate double-stranded cDNAs; here random primers are used to initiate cDNA synthesis, the products of which are then stitched together by DNA ligase. Note that since priming of second strand synthesis occurs randomly, this may not always begin from the very end of the first strand template, as depicted in the example here. Adaptor–motor complexes are then ligated to the double-stranded cDNA ends, although in instances where the first strand overhang might be particularly long, as in the example here, it is unlikely that the adaptor–motor complex will ligate efficiently to enable sequencing of the second strand. **b** To better enrich for full-length cDNAs, the ONT-DCS108 direct cDNA sequencing kit, which leverages template-switching[17], was used. When the first strand cDNA synthesis reaches the end of the RNA molecule, the reverse transcriptase will add a few non-templated Cs to the end of the cDNA. A strand-switching primer (SSP) present in the reaction binds to these non-templated Cs, and the reverse transcriptase then switches template from the RNA to the SSP. The second cDNA strand, presuming its synthesis continues to the end of the first strand template as in the example here, will also span the full length of the primary polyA RNA template; note, however, that in many instances a non-full-length second cDNA strand will likely be sequenced. **c** The ONT-RNA001 workflow enables sequencing of native RNA strands. Here an oligodT-adaptor-motor complex is ligated to the polyA end of the RNA. In order to relax the secondary structure of the RNA (and thus help ensure efficient translocation of the RNA strand through the nanopore), a cDNA synthesis step is performed. Since only the RNA strand has a motor ligated, the RNA molecule, but not the cDNA strand, is sequenced

disambiguation would require a reliable model of the read generation process, accounting for the probability of transcript truncation in the library preparation step and/or read truncation during the sequencing-basecalling process. To investigate to what extent the secondary alignments in our libraries could be the result of nested sets of reference transcripts, we extracted all reads with at least one secondary transcriptome alignment, and among all primary and secondary alignments, we selected the one for which the covered portion of the target transcript by the read was highest. If the secondary alignments are the result of the true transcript of origin being contained in the other target transcripts, we expect this maximally covered portion to be close to 1. While we did notice a clear peak close to 1 for most data sets, there was also a broad distribution of lower coverage degrees (Supplementary Fig. 8). Taken together, these observations suggest that, despite the long-read length, unambiguously inferring the true transcript of origin for any given read is still highly nontrivial, and simply selecting the reported primary transcriptome alignment for downstream analysis can give misleading results.

While secondary alignments represent possible mapping positions of a read beyond the one reported as primary, supplementary alignments arise when a read cannot be mapped in a contiguous fashion, and consequently minimap2 splits the alignment into multiple parts. We observed a comparatively large number of supplementary alignments in the ONT-DCS108-HAP data set, both for genome and transcriptome alignments (Fig. 2b). Further investigation revealed that in this data set, as well as in ONT-NSK007-HAP, a relatively large fraction of the supplementary alignments overlapped the corresponding primary alignment, but on the opposite strand (Fig. 2c). In these cases, the

overlap between the primary and supplementary alignments was often large (Fig. 2d). There was also an enrichment of reads containing long palindromes (i.e., a sequence as well as its perfect reverse complement) in the ONT-DCS108-HAP data set compared with ONT-RNA001-HAP, and additionally an enrichment of such sequences among reads with reported supplementary alignments (Supplementary Fig. 9). These observations are interesting, as we note that the ONT 1D[2] sequencing mode (https://nanoporetech.com/) exploits the observation that the second strand of a double-stranded DNA molecule often enters the nanopore immediately following the first strand during 1D sequencing. 1D[2] sequencing chemistry is designed to further promote this observed phenomenon, and the associated 1D[2] basecaller is customized to efficiently split reads according to each strand sequenced. Thus, observations of frequent overlapping primary-supplementary alignments on opposite strands during 1D cDNA sequencing may reflect un-split reads by the standard 1D basecaller.

A peak of short low-quality unfiltered reads was consistently observed in the native RNA libraries (Supplementary Fig. 1B), and the majority of these did not align adequately to either the genome or the transcriptome (Supplementary Fig. 5A, B). More generally, for aligned reads, in particular those shorter than 10,000 bases, most of the individual bases could be matched to a position in the reference sequence, indicated by a large fraction of Ms and consequently a low fraction of insertions, deletions and soft-clipped bases in the CIGAR string (Fig. 2e, Supplementary Figs 5, 10). Reads longer than 10,000 bases, which were mostly found in the cDNA libraries, typically did not align end-to-end (Fig. 2e). For the ONT-DCS108-HAP libraries, a large fraction of

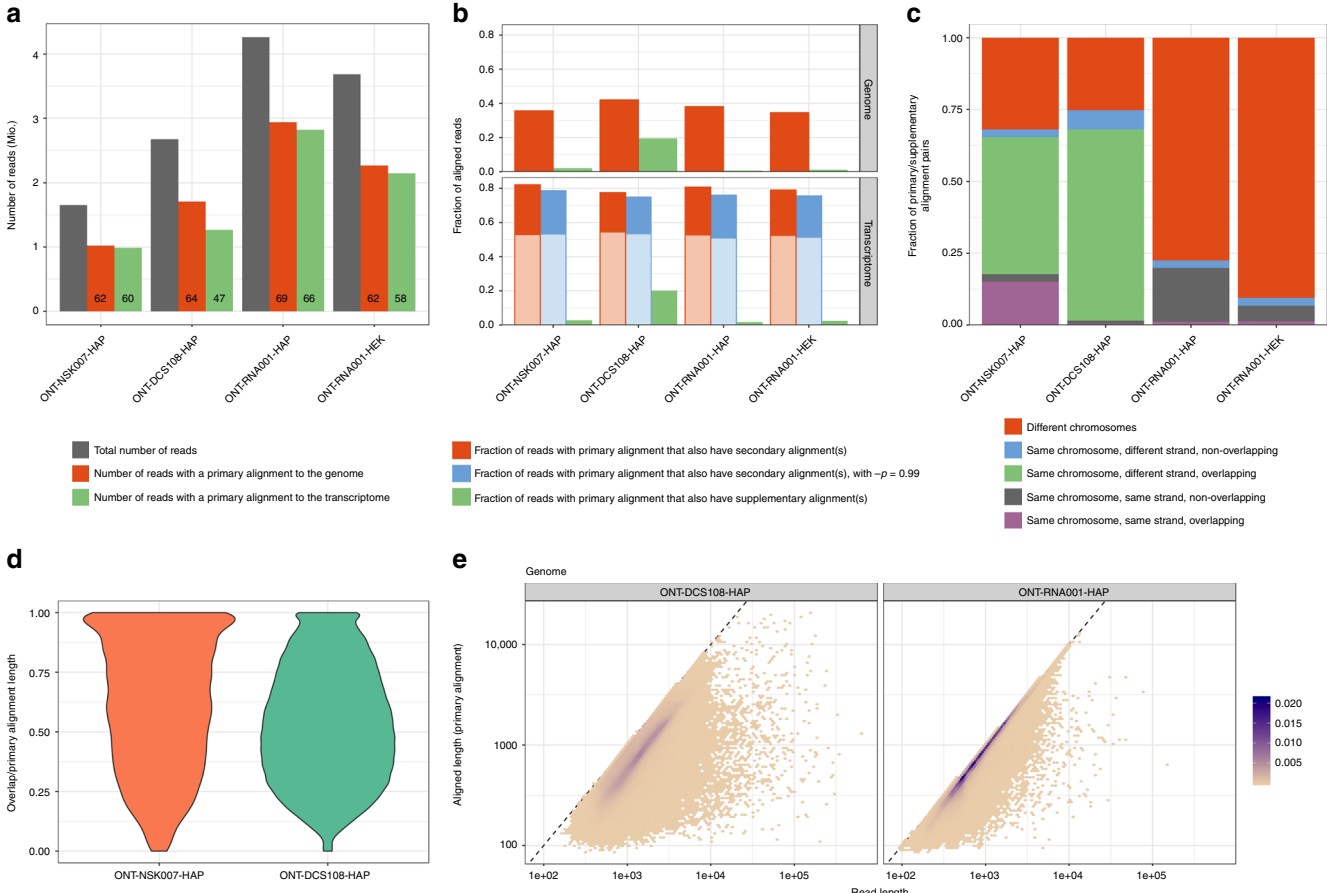

**Fig. 2** Characterization of aligned reads. **a** Total number of reads and number of reads with a primary alignment to the genome and transcriptome, respectively, in each of the ONT data sets. The number displayed in each bar represents the alignment rate in % (the fraction of the total number of reads for which minimap2 reports an alignment). **b** Fraction of the reads with a primary alignment to the genome or transcriptome, respectively, that also have at least one reported secondary or supplementary alignment. The lighter shaded parts of the secondary transcriptome alignment bars correspond to reads where all primary and secondary alignments are to isoforms of the same gene, while the darker shaded parts correspond to reads with reported alignments to transcripts from different genes. **c** Investigation of supplementary genome alignments. Each supplementary alignment is categorized based on whether it is on the same chromosome and strand as the primary alignment, and if the alignment positions of the primary and supplementary alignments overlap. **d** Length of the overlap between the primary and supplementary alignments, divided by the primary alignment length (number of M and D characters in the CIGAR string), for supplementary alignments falling on the same chromosome, but different strand, and overlapping, the primary alignment (light green fractions in panel **c**). **e** Total read length ($x$) vs aligned length ($y$, the sum of the number of M and I characters in the CIGAR string) for the primary genome alignment of each read, summarized across the replicates in the ONT-DCS108-HAP and ONT-RNA001-HAP data sets. The color indicates point density. Source data for panels **a**–**c** are provided as a Source Data file

the bases in the primary alignments were soft-clipped, corresponding to the large number of supplementary alignments discussed above. Incorporating the genomic coordinates of the annotated genes, we also observed differences in the gene body read coverage distribution between the libraries (Supplementary Fig. 11), with a stronger 3′ coverage bias in the cDNA libraries than in the native RNA libraries.

**Coverage of full-length transcripts by individual ONT reads.** To investigate to what extent individual ONT reads can be expected to represent full-length transcripts, we selected the best target transcript for each read as described in Methods. As expected, since the ONT-NSK007-HAP library preparation does not involve full-length cDNA enrichment (Fig. 1a), these reads achieved a much lower degree of full-length transcript coverage across the range of transcript lengths (Fig. 3a). In contrast, transcripts shorter than 2 kb could often be completely covered by a single read in the other libraries, although this was more rarely the case for longer transcripts (Fig. 3a, Supplementary

Fig. 12). Recently, a preprint described the analysis of 9.9 million aligned native RNA reads from the NA12878 human reference cell line[18], and reported the presence of frequently truncated reads when sequenced mitochondrial transcripts were analyzed. We therefore next investigated whether read truncation might be a more general feature also of the NA12878 data set. While there appeared to be some more subtle differences between the native RNA sequencing output from the three different cell lines (Fig. 3a: ONT-RNA001-HAP and ONT-RNA001-HEK, Fig. 3b: NA12878), the overall observations and trends were similar, with transcripts longer than 2 kb particularly often poorly covered by a single read. Applying the same procedure to the SIRV and ERCC data sets from Garalde et al.[15] revealed that a majority of these synthetic transcripts were well covered by single reads (Fig. 3c, d), confirming observations from previous studies[9,15]; importantly, however, all transcripts in the SIRV and ERCC catalogs are shorter than 2.5 kb, while ~17% of the transcripts in the Ensembl GRCh38.90 catalog are longer than that. This suggests that while the synthetic transcript catalogs provide useful information about the performance of long-read

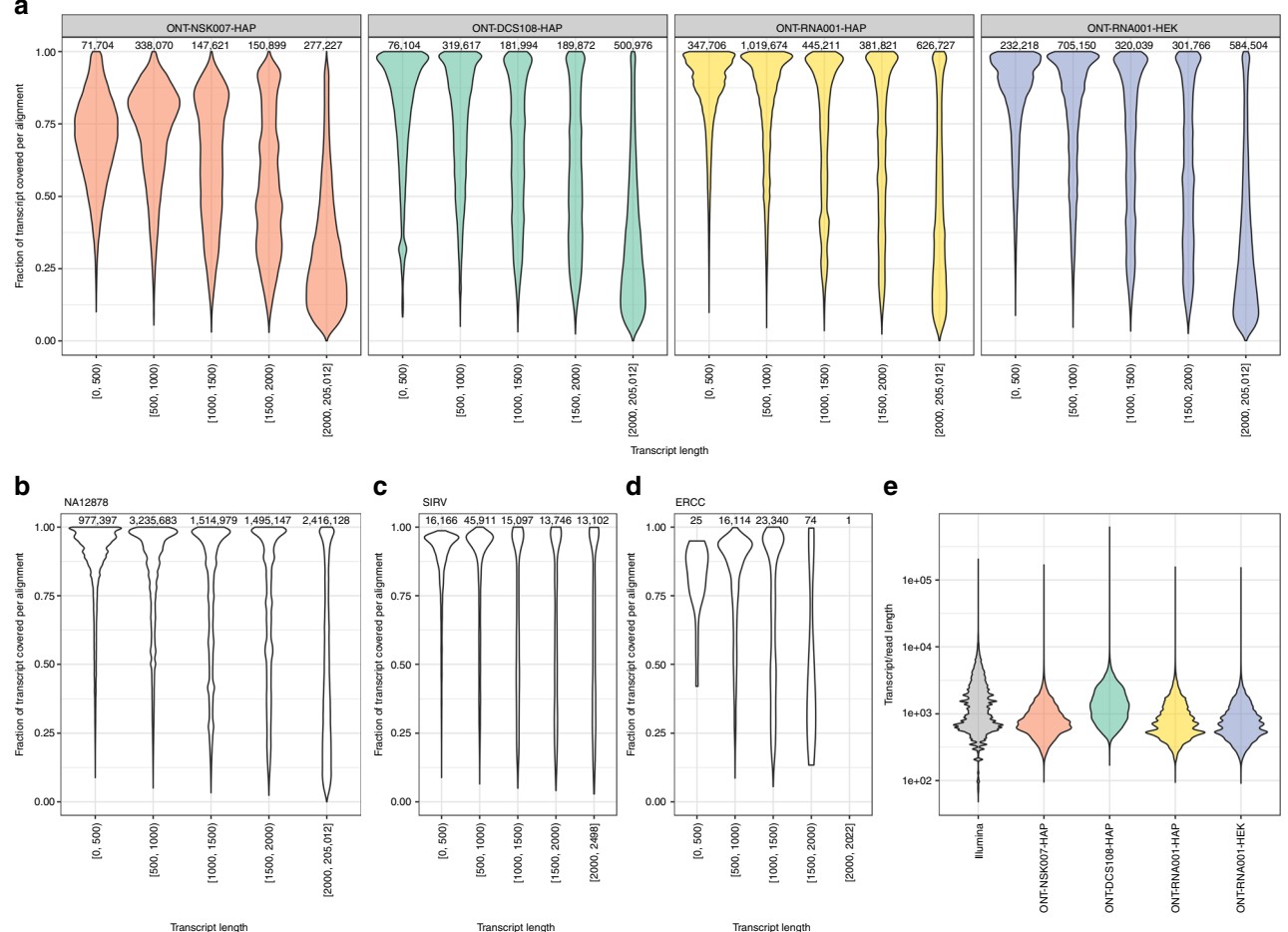

**Fig. 3** Transcript coverage fraction by individual reads. **a** Distribution of coverage fractions of transcripts by individual reads, for each of the four ONT data sets, stratified by the length of the target transcript. The target transcript was selected to maximize the coverage fraction, among all reported long enough alignments (see Methods), and thus the reported coverage fractions represent upper bounds of the true ones. The number above each violin indicates the number of processed alignments to transcripts in the corresponding length category. **b–d** Distribution of coverage fractions of transcripts by individual reads for the NA12878, SIRV, and ERCC data sets. **e** Observed distribution of raw read lengths for reads with at least one genome alignment (for ONT data sets) and expected distribution of transcript molecule lengths based on annotated transcript lengths and estimated abundances in the Illumina samples

transcriptome sequencing and analysis methods, extrapolation of the results to real, complex transcriptomes should be done with care.

To further investigate the degree to which individual ONT reads are likely to represent full-length transcripts, we compared the observed raw ONT read length distribution (for reads with at least one genome alignment) with the expected transcript length distribution in these samples, obtained by weighting the annotated transcript lengths by the estimated transcript abundances (in transcripts per million) estimated by Salmon in the Illumina samples. This analysis showed an apparent shortage of ONT reads in the length range of the longest transcripts inferred to be expressed in the Illumina data (Fig. 3e). The ONT-DCS108-HAP samples were the exception; however, for many of the reads in these libraries, the primary alignment does not cover the entire read (Fig. 2e, Supplementary Fig. 13A–C). Further inspection of the longer annotated transcripts with high estimated abundances in the Illumina samples, the majority of which were protein coding, revealed consistent base pair coverage by Illumina reads along their length (Supplementary Fig. 13D), indicating that they were indeed truly present. Overall, such observations further illustrate that using current library preparation and sequencing workflows, long transcripts are often not represented by single ONT reads.

**Reference-based transcript detection and quantification.** Four reference-based methods were used to estimate transcript and gene abundances in each of the ONT libraries. For two of these methods, we specifically evaluated the impact of data preprocessing: for minimap2 followed by Salmon in alignment-based mode (denoted salmonminimap2), we investigated the effect of setting the -p argument of minimap2 to different values (the default of 0.8 as well as 0.99) in the transcriptome alignment step, and for Salmon in quasi-mapping mode, we evaluated the effect of providing only the aligned bases of the reads with a primary alignment anywhere in the genome (see Methods). Increasing -p to 0.99 led to a slightly improved correlation between ONT transcript read counts and estimated transcript abundances from the Illumina samples (obtained by Salmon in quasi-mapping mode), and thus, in the following analyses, we set -p equal to 0.99 for Salmon following minimap2 (Supplementary Fig. 14). Removing the non-aligned bases before running Salmon did not improve the correlations notably (Supplementary Fig. 14). Since this is a more involved procedure, and further introduces a dependency on the genome alignments, we use the Salmon quantifications obtained using the original, non-truncated reads for the rest of the analyses.

We observed a large difference between the numbers of reads assigned to features by the different methods (Supplementary

Fig. 15). The highest assignment rates were consistently obtained with salmonminimap2, where all reads that were aligned to the transcriptome were also subsequently assigned to features. featureCounts assigned a slightly lower fraction of the reads to genes, while Salmon in quasi-mapping mode and Wub assigned considerably fewer reads. However, the relatively low number of reads assigned by Salmon in quasi-mapping mode were

distributed across as many, sometimes more, genes and transcripts as the reads assigned by salmonminimap2 (Fig. 4a, b), suggesting that no category of genes or transcripts was consistently missed. In general, the transcript-level detection rate increased with transcript length, both for ONT and Illumina libraries (Fig. 4c). Counting the number of detected transcripts and genes at various degrees of subsampling (Fig. 4d, e) suggested

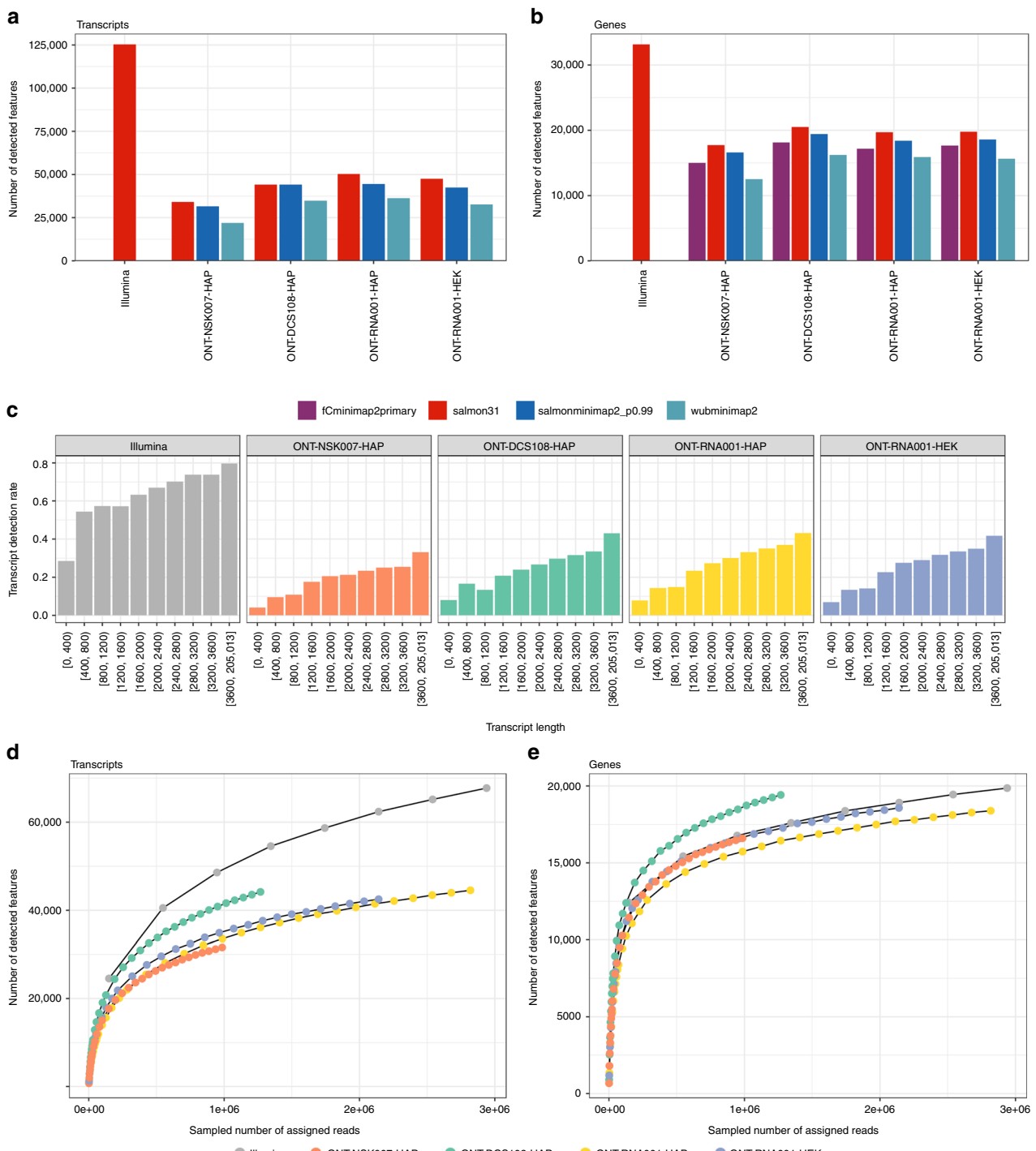

**Fig. 4** Detection of annotated transcripts and genes. **a**, **b** Number of detected transcripts and genes with the applied abundance estimation methods. Here, a feature is considered detected if the estimated read count is ≥1. **c** Fraction of transcripts detected (with estimated count ≥ 1), stratified by transcript length, in the respective data sets. **d**, **e** Saturation of transcript and gene detection in ONT and Illumina data sets. For each data set, we aggregated reads across all replicates, subsampled the reads and recorded the number of transcripts and genes detected with an estimated salmonminimap2 count (ONT libraries) or Salmon count (Illumina libraries) ≥1. The Illumina curves are truncated to the range of read numbers observed in the ONT data sets. Source data are provided as a Source Data file

that the current sequencing depth of up to 3 million mapped ONT reads per library type was not enough to detect all expressed genes or transcripts. Furthermore, the number of detected genes were similar to the number observed in the Illumina libraries if these were subsampled to comparable sequencing depths. With the aim of investigating whether there are systematic blind spots in the detection of features in the ONT data (in which case we expect the same set of transcripts to be detected in all libraries) or if the lack of saturation is purely a result of undersampling (in which case we would expect differences in the set of detected transcripts across libraries), we compared the saturation curves obtained from individual samples to that obtained by first pooling the reads across all replicates within a data set, and subsequently sampling from this pool (Supplementary Fig. 16). On the transcript level, pooling the samples improved the degree of saturation for a given number of reads, while no improvement could be seen on the gene level.

Next, we calculated the correlation between abundance estimates among replicates of the HAP cell line, within and between data sets. As expected, the correlation between replicates was higher on the gene level than on the transcript level, and higher within a data set than between data sets (Supplementary Fig. 17). On the gene level, correlation between replicates was almost as high in the ONT data as in the Illumina data, for all

quantification methods, while for transcript-level abundances, higher correlations were observed in the Illumina data. Overall, Wub showed the highest correlation of abundance estimates between replicates in the ONT data sets. Notably, correlations between cDNA and native RNA samples were often as high as those among samples obtained with different cDNA protocols.

Comparing the abundance estimates obtained for the same library with different quantification methods showed that, perhaps unsurprisingly, Salmon in quasi-mapping mode and salmonminimap2 had the highest correlation (Supplementary Fig. 18). Stratifying transcripts and genes by the annotated biotype suggested that certain biotypes (in particular, short transcripts such as miRNAs) were consistently assigned very low abundances with ONT, while they were considered expressed by Salmon in the Illumina libraries (Supplementary Fig. 19).

**Transcript identifiability**. Next, we focused on specific transcriptomic features that are useful for discriminating between similar isoforms. First, we extracted the junctions observed after aligning the ONT reads to the genome. The majority of the junctions that were covered by at least five ONT reads were already annotated in the reference transcriptome, while this was more rarely the case for lowly covered junctions (Fig. 5a, b,

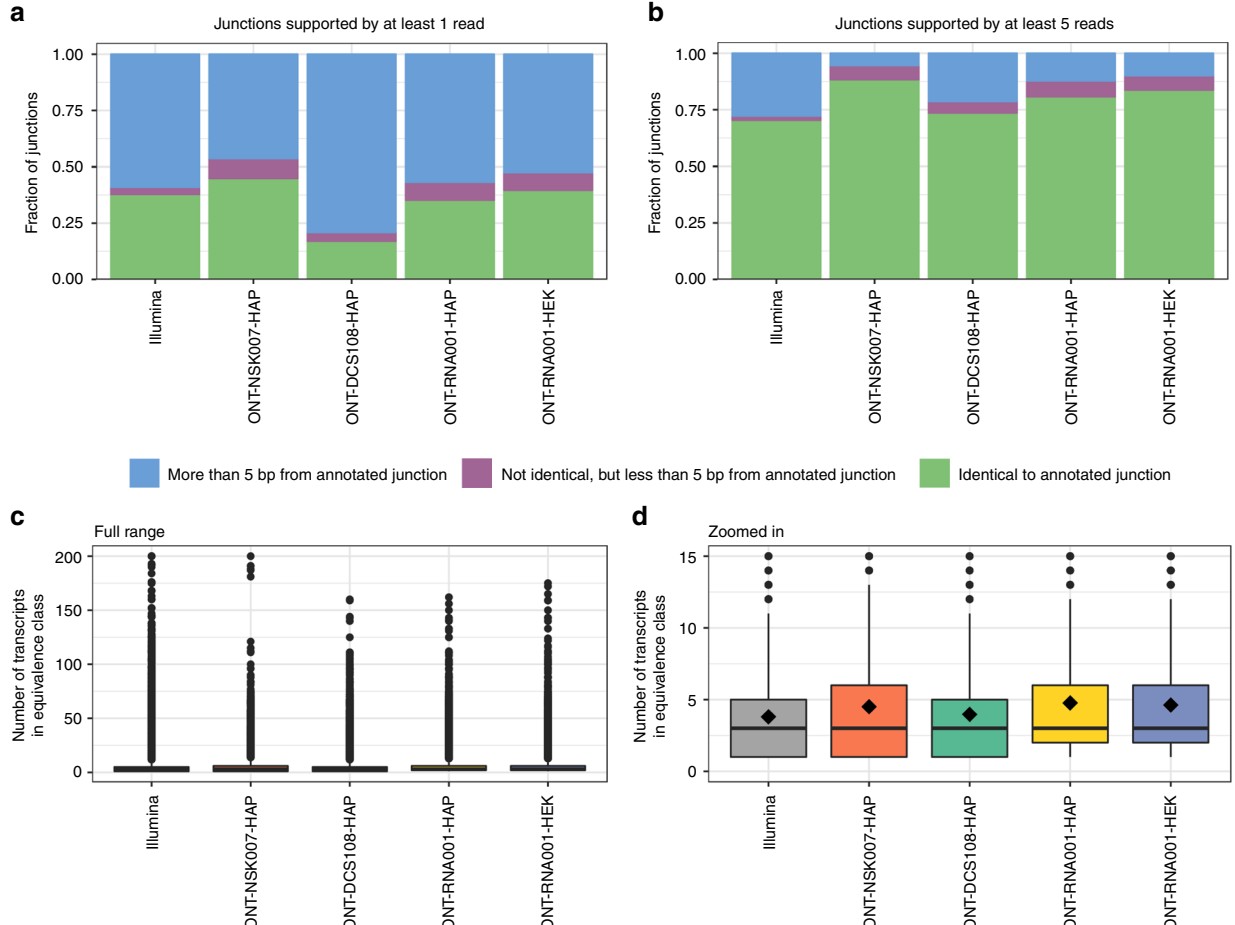

**Fig. 5** Investigation of transcript identifiability. **a**, **b** Annotation status of junctions observed in each ONT and Illumina data set. A junction is considered observed if it is supported by at least 1 (**a**) or 5 (**b**) reads. For each observed junction, the distance to each annotated junction was defined as the absolute difference between their start positions plus the absolute difference between their end positions. This distance was used to find the closest annotated junction. **c** Distribution of the number of transcripts contained in the Salmon equivalence class that a read is assigned to, across all reads, for each ONT and Illumina data set. The center line represents the median; hinges represent first and third quartiles; whiskers the most extreme values within 1.5 interquartile range from the box. **d** As (**c**), but zoomed in to the range [0, 15]. The black diamond shape indicates the mean. Source data for panels **a**, **b** are provided as a Source Data file

Supplementary Fig. 20A, B). Junctions that were observed in the ONT reads but did not correspond to annotated junctions were less likely than those already annotated to be observed in the Illumina data, and also less likely to harbor a canonical splice junction motif (GT-AG) (Supplementary Fig. 21). Not surprisingly, individual ONT reads generally spanned more junctions than Illumina reads (Supplementary Fig. 22), which should provide improved ability of correct transcript identification.

In order to further investigate if the increased length of ONT reads improved their unambiguous assignment to specific transcripts, we tabulated the number of transcripts included in the equivalence class that each read was assigned to when running Salmon in quasi-mapping mode. A read being assigned to a large equivalence class indicates that the read sequence is compatible with many annotated transcripts, and consequently that unambiguous assignment is difficult. While fewer ONT reads were assigned to equivalence classes with a very large number of transcripts compared with the Illumina counterparts, the average number of transcripts in the equivalence class, across all reads, was almost identical (Fig. 5c, d, Supplementary Fig. 20C, D). To investigate to what extent this was an effect of the high redundancy among the annotated transcripts, we ran Salmon with the same index, but using the annotated transcript catalog as a proxy for error-free, full-length reads. In this case, 87% of these read proxies were assigned to single-transcript equivalence classes. This illustrates both that even in this idealized situation, not all reads would be unambiguously assignable to a single annotated isoform, and that for the real ONT reads, the ambiguity is still considerably higher than in the ideal situation. Together with the large number of secondary transcriptome alignments observed above, this illustrates the challenging nature of reference-based transcript identification based on ONT reads.

**Reference-free transcript identification**. In addition to the reference-based transcript identification and quantification discussed above, we generated a set of high-confidence consensus transcripts for each ONT data set using FLAIR (https://github.com/BrooksLabUCSC/flair). The identified transcripts were compared with the annotated reference transcriptome using SQANTI[19] and gffcompare (https://ccb.jhu.edu/software/stringtie/gffcompare.shtml), assigning to each transcript a structural category (SQANTI) or class code (gffcompare), describing the type of relationship to the most similar reference transcript. Only a relatively low fraction of the identified transcripts in each data set contained a junction chain that was identical to that of an annotated transcript (Fig. 6a, structural category 'full-splice_match', Supplementary Fig. 23, class code ' = '), with an additional fraction of the identified transcripts having a junction chain that was consistent with an annotated transcript, but only contained a subset of the junctions. This corroborates the previous observations that many ONT reads may not represent full-length transcript sequences. There is a marked difference compared with the set of transcripts assembled with StringTie from the Illumina samples, a larger fraction of which contain a complete intron chain match with an annotated transcript. There is also a larger fraction of Illumina-derived transcripts that do not overlap known transcripts (Fig. 6a, structural category 'intergenic', Supplementary Figs. 23, class code 'u'). FLAIR transcripts with a junction chain perfectly matching an annotated transcript spanned a range of lengths and number of junctions (Fig. 6b, c, Supplementary Figs. 23, 24), suggesting that transcript identification is not limited to, e.g., short isoforms. Overall, the set of transcripts assembled by StringTie from the Illumina data were more often multi-exonic than those from the ONT libraries, and

also spanned a broader range of transcript lengths. Supplying the set of splice junctions seen in the short-read libraries to FLAIR when identifying transcripts generally led to a larger number of transcripts (Fig. 6a) and, among those, a larger fraction of novel transcripts with splice junctions that were not included in the reference catalog. While the categories used by SQANTI and gffcompare are different, there is often a clear association between them (Supplementary Fig. 25). A random selection of FLAIR transcript sequences (from the ONT-RNA001-HAP library) corresponding to annotated transcripts are shown in Supplementary Fig. 26, to illustrate the variety of transcripts that could be identified.

Comparing the set of annotated reference transcripts that could be identified by at least one FLAIR transcript (SQANTI structural category 'full-splice_match' or 'incomplete-splice_match') in the respective ONT data sets showed that a large fraction of these transcripts were only identified in a single data set (Fig. 7a), while others were only identified if Illumina junctions were supplied to FLAIR at runtime. In addition, reference transcripts identified by the native RNA sequencing protocol in the two different cell lines showed a high degree of similarity to each other, suggesting that transcript identification can be strongly affected by the library preparation protocol. Of note, the native RNA protocols provide information about the strandedness of the reads, which is not the case for the cDNA protocols employed here. Reference transcripts with junction chains corresponding to at least one FLAIR transcript generally showed a higher expression level in the Illumina samples than the reference transcripts that were not identified in any ONT data set (Fig. 7b), suggesting that one possible explanation for the discrepancy between the transcripts identified in the different ONT data sets could be the limited sequencing depth, and that a larger number of ONT reads may be necessary to identify a stable set of expressed transcripts.

**Investigation of polyA tail length**. Eukaryotic mRNAs can harbor varying lengths of polyA tails at their 3′ termini, which can regulate mRNA turnover, as well as influence several other biologically important phenomena[20]. Thus, whereas a more standard polyadenylation takes place co-transcriptionally in the nucleus, cytoplasmic polyadenylation of mRNA transcripts has been shown to regulate diverse processes such as synaptic plasticity, oocyte maturation, and circadian rhythm/biological timing[20]. Accordingly, important sequencing-based techniques have been introduced to enable measurement of polyA length, most notably the TAIL-seq method[21]. While the implementation of TAIL-seq has improved our understanding of polyA tail length function to a degree, the relatively laborious technical nature of the procedure has likely limited its more widespread use. Thus, although the main focus of our study was on the ability to accurately estimate transcript and gene abundance levels and identify expressed features from ONT reads, we also performed some analysis into characterizing the length of transcript polyA tails from Nanopore data. We investigated native RNA reads from an ONT-RNA001-HAP library using Nanopolish[18] and tailfindr[22], which displayed largely concordant estimates of polyA length (Fig. 8a, b), and further appeared in agreement with a modal polyA length of ~50–100 nt observed via TAIL-seq experiments from mammalian cell lines[21]. Furthermore, we were able to observe differential polyA tail length distributions between various RNA biotypes, as well between subtypes of protein coding genes (Fig. 8c). Most notably, transcripts from all mitochondrial gene subtypes, including mitochondrial protein coding genes, displayed generally shorter polyA tail lengths, in accordance with previous independent observations[18,23].

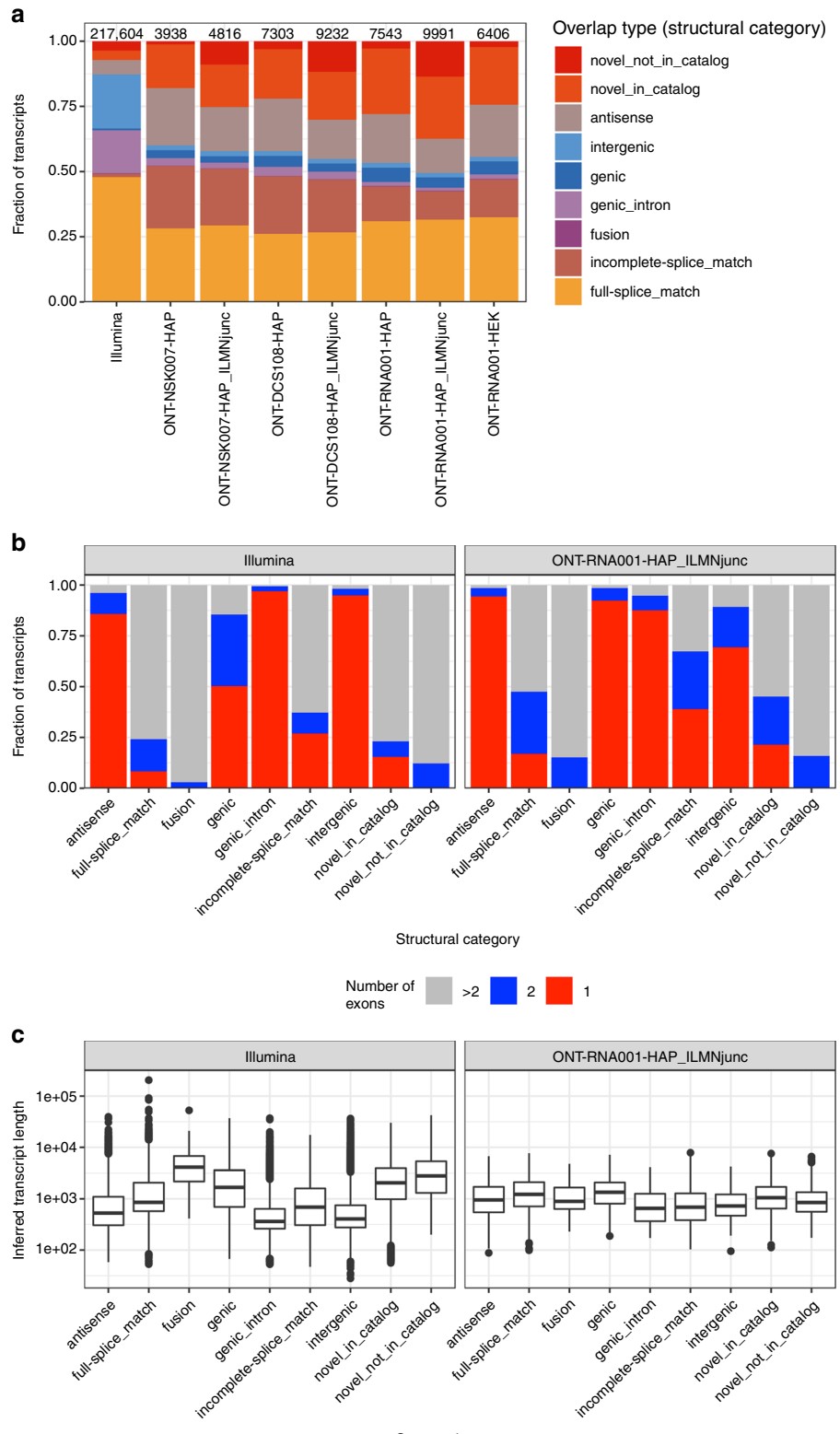

**Fig. 6** Characterization of transcripts identified by FLAIR. **a** Structural category distribution for de novo identified transcripts from FLAIR (for ONT libraries) or StringTie (for Illumina libraries), compared with the set of annotated transcripts using SQANTI. The number above each bar represents the number of assembled transcripts. The structural category for a transcript indicates its relation to the closest annotated transcript. The _ILMNjunc suffix indicates that junctions identified in the Illumina libraries were supplied when running FLAIR. **b** Number of exons in each transcript identified by FLAIR/StringTie, stratified by the relation to the annotated transcripts (represented by the assigned structural category). **c** Length distribution of transcripts identified by FLAIR/StringTie, stratified by the relation to the annotated transcripts (represented by the assigned structural category). The center line represents the median; hinges represent first and third quartiles; whiskers the most extreme values within 1.5 interquartile range from the box. Source data for panels **a**, **b** are provided as a Source Data file

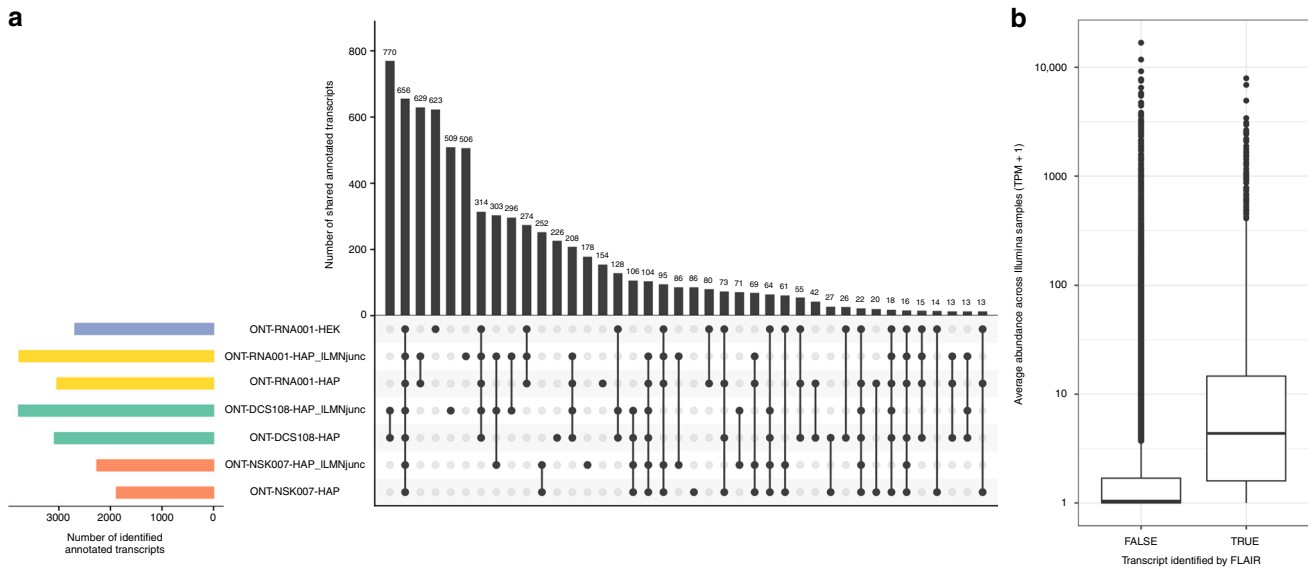

**Fig. 7** Comparison of annotated transcripts identified by FLAIR in the four ONT data sets. **a** UpSet plot representing overlaps between the annotated transcripts that are identified by FLAIR in the different ONT data sets. An annotated transcript is considered to be identified if at least one FLAIR transcript is assigned to it with a structural category annotation of either 'full-splice_match' or 'incomplete-splice_match'. These sets of annotated transcripts are then compared between data sets. Horizontal bars indicate the total number of identified annotated transcripts in the respective data sets, and vertical bars represent the size of each intersection of one or more sets of identified transcripts. **b** Average abundance across the Illumina samples, for annotated transcripts that are considered identified or not by FLAIR. An annotated transcript is considered identified if at least one FLAIR transcript from at least one data set is assigned to it with a structural category annotation of either 'full-splice_match' or 'incomplete-splice_match'. The center line represents the median; hinges represent first and third quartiles; whiskers the most extreme values within 1.5 interquartile range from the box

## Discussion

We have performed a detailed evaluation of reads from Nanopore native human RNA sequencing as well as complementary direct cDNA sequencing, from the perspective of transcript identification and quantification. We observed that despite the fact that ONT reads are around an order of magnitude longer than typical Illumina reads, identification of their transcript of origin is still highly nontrivial, and a large number of secondary transcriptome alignments with mapping scores very close to the primary alignments were observed for all libraries. This suggests that quantification methods that focus exclusively on the reported primary alignment are likely to be suboptimal, and can be highly biased depending on how the primary alignment is selected among a set of equally good mappings. We expect that reference-based transcript abundance estimation methods that are able to incorporate information about these multimapping reads are more likely to produce reliable abundance estimates; however, to our knowledge no such ONT-specific method, with a read generation model adapted to the ONT data characteristics, currently exists.

De novo as well as reference-based identification of transcripts suggested that a considerable number of the raw ONT reads are unlikely to represent full-length reference transcripts. This can have implications for transcript identification and quantification. For example, it is difficult to determine whether a truly truncated version of a reference transcript is present in a sample, or if the reads rather are fragments of longer transcript molecules. In addition, by attempting to mitigate this issue, e.g., by filtering the ONT reads to only retain those that overlap a known promoter region, the quantitative nature of the data, as well as the number of usable reads, may be reduced. The overall causes for the observed direct cDNA and native RNA truncations are unclear, but could well vary, with the former, for example, potentially influenced by the suboptimal nature of template-switching to directly select for full-length cDNAs during library preparation, and the latter more specifically influenced by factors during the

sequencing process itself. For example, the Nanopore RNA-seq consortium study estimated that a significant proportion of native RNA transcripts may be truncated by nanopore signal noise caused by electrical signals associated with RNA motor enzyme stalls, or by otherwise stray current spikes of unknown origin[18]. We also agree that nanopore native RNA read truncation is unlikely due to some fundamental limitation of nanopore-based sequencing, especially considering that ONT 1D genomic DNA sequence reads of several kilobases are consistently achieved without issue using the current pore type[16,24,25] used to sequence both DNA and RNA. Further, such problems could conceivably be addressed, to at least some extent, by training basecallers to reliably recognize relevant nanopore signal noise events which might cause single molecule sequence reads to be truncated or split.

Our observations of ONT native RNA sequencing-mediated polyA tail length measurement were encouraging, and corroborated results from independent analysis methods as well as past observations made using orthogonal techniques. Considering its relative methodological simplicity in terms of library preparation and sequencing workflow, we suggest that native RNA-seq along with the available analysis tools, Nanopolish[18] and tailfindr[22], might already be considered state-of-the-art for more routine transcriptomic investigations of polyA tail lengths. One of the additional central advantages that ONT native RNA sequencing offers, that we have not investigated here, is the potential for direct detection of RNA ribonucleotide modifications, which are known to be key regulators of a wide range of practically important aspects of biology[26]. Current sequencing-based epi-transcriptomics methods rely on sequencing via a DNA intermediate in order to infer the RNA modification site; such approaches are, however, often not ideal as they can yield high numbers of inaccurate modification calls[27]. The development of accurate and robust non-canonical basecalling tools for ONT native RNA-seq data would potentially solve many of the unresolved issues in the field.

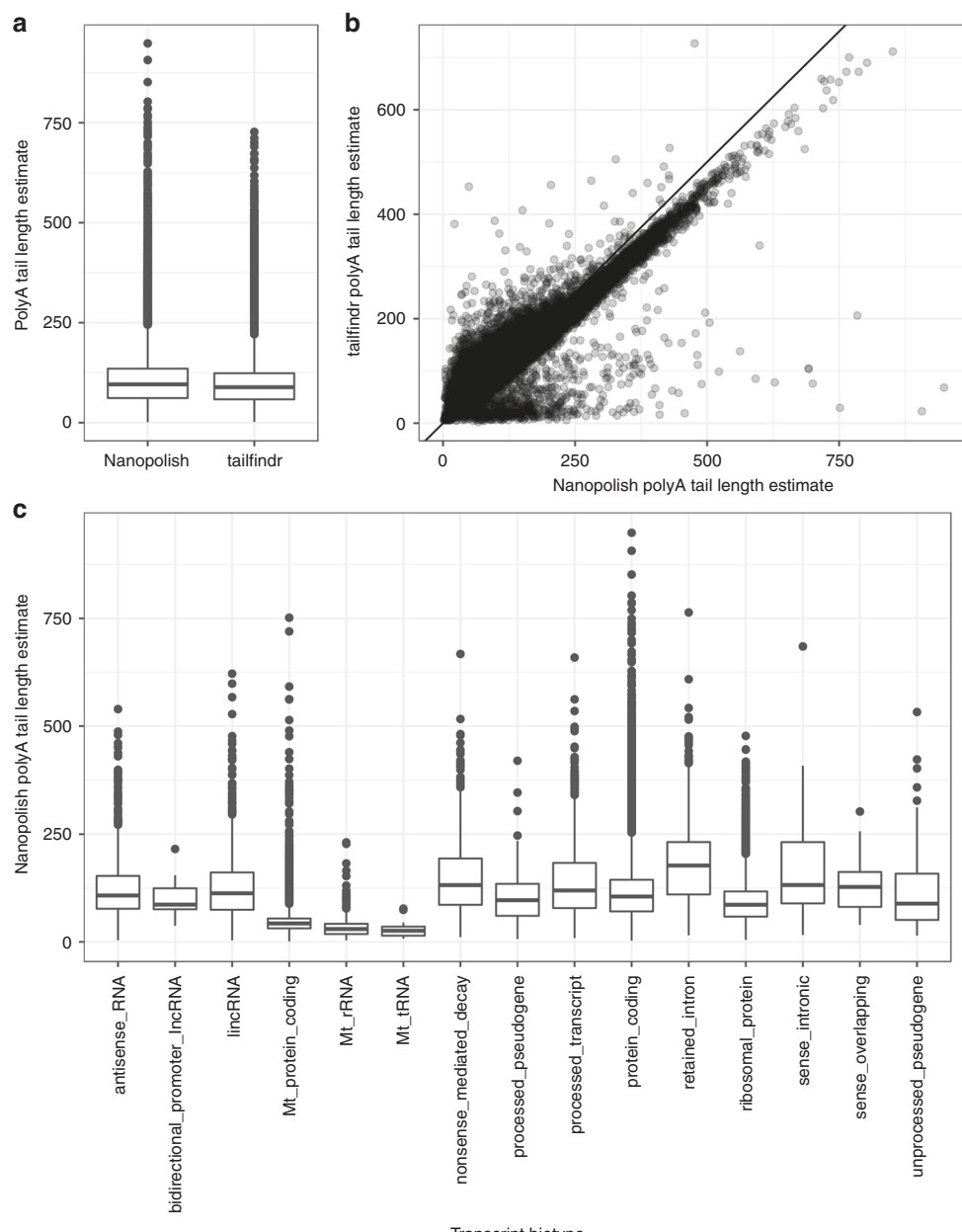

**Fig. 8** Evaluation of polyA tail length estimates. **a** Overall distribution of polyA tail length estimates from Nanopolish and tailfindr. **b** Agreement between polyA tail length estimates from Nanopolish (*x*) and tailfindr (*y*). **c** Distribution of estimated polyA tail lengths for reads assigned to transcripts of different biotypes. Only biotypes with at least 10 assigned reads are shown. For boxplots, the center line represents the median; hinges represent first and third quartiles; whiskers the most extreme values within 1.5 interquartile range from the box. In these plots, a single outlier read with a polyA tail length estimate from Nanopolish exceeding 3 kb was excluded

An inability to read ~10–15 nucleotides at the 5′ end of each strand, and relatively higher error rates, were identified as the two principal drawbacks of native RNA sequencing by the Nanopore RNA-seq consortium study, although these are potentially readily addressable[18]. Here we highlight that the sequencing depths achieved from native RNA libraries from single flow cells (typically ~0.5 M aligned reads) are likely not enough to saturate transcript detection, either using reference-based or de novo approaches. Improving throughput (the amount of sequence rendered per unit cost and unit time) is a critical issue; although protein-pore sequencing can be scaled to considerably higher levels (i.e., either on the ONT GridION or PromethION instruments), the associated consumable nanopore array costs remain high. Thus, native RNA-seq throughput characteristics that are

deemed acceptable by the transcriptomics community at large will likely require a highly optimized RNA motor enzyme, or ultimately a shift to a lower cost or more durable nanopore array type. When characterization of complex transcriptomes at transcript-level comprises the project remit, our study here describes that Nanopore direct RNA-seq remains a roundly promising but fledgling analysis tool.

## Methods

**Cell lines and culture**. HEK293 cells (ATCC, catalog# CRL-1573) were cultured in Dulbecco's modified Eagles medium (DMEM) supplemented with 10% FBS and penicillin/streptomycin. HAP1 cells (Horizon Discovery, catalog# C631) were grown in Iscove's modified Dulbecco's medium (IMDM) supplemented with 10% FBS and penicillin/streptomycin. All cultures were maintained at a temperature of 37 °C in a humidified incubator with 5% $CO_2$. When required, exponentially

growing cells were harvested by washing in phosphate buffered saline (PBS) and then incubating with Trypsin-EDTA, followed by further washing of pelleted cells in PBS.

**Library preparation and sequencing.** For the Nanopore libraries, total RNA was extracted from cell pellets using Trizol, and the DNase-treated samples were then polyA-selected using oligodT dynabeads (Invitrogen). The ONT kits NSK007, DCS108, and RNA001 were then used for PCR-free 1D library preparations. For each RNA001 library (6× HAP & 5× HEK), 500 ng of input polyA + RNA was used, and these were made following ONT instructions. For DCS108, 100 ng of input polyA + RNA was used per library (2× HAP), and these were prepared according to the ONT-recommended protocol. For NSK007 libraries (2× HAP), 100 ng of input polyA + RNA was used, and these were made according to ONT instructions, except that the hairpin adaptor (HPA) ligation and PCR steps were omitted in order to enable 1D direct cDNA sequencing. The 15 prepared libraries were sequenced on the MinION using R9.4 flow cells with the relevant MinKNOW script to generate fast5 files. All generated fast5 reads were then basecalled in Albacore (version 1.2.6 for NSK007 libraries and version 2.1.0 for DCS108 and RNA001 libraries) using the relevant script to yield fastq files. As Albacore only contained a 2D script for NSK007 basecalling, only the generated NSK007 fastq raw reads (i.e., complement and template) were taken forward for analysis, while any attempted consensus reads present were discarded. We noted concerns of previous studies reporting that filtering of reads during basecalling often resulted in a significant number of useful good-quality reads being discarded[10]. Indeed, in later versions of available ONT Albacore packages, filtering was either turned off as default, or its disabling offered as an option. As sequencing depth would likely be the key limiting factor influencing our downstream analyses, and reasoning that true low-quality reads would be filtered during the alignment step, we thus made use of the Albacore non-filtering option for all sequenced libraries.

For the Illumina samples, all libraries were made using the Illumina TruSeq stranded mRNA kit. The mRNA libraries were prepared from 500 ng of Trizol-extracted total RNA using the Illumina TruSeq® Stranded mRNA Sample Preparation Kit with 15 PCR cycles applied. Libraries were quantified and quality checked using qPCR with Illumina adapter specific primers and Agilent 2200 TapeStation, respectively. Diluted indexed mRNA-seq (10 nM) libraries were pooled, used for cluster generation (Illumina TruSeq PE Cluster Kit v4-cBot-HS) and sequenced [Illumina HiSeq 4000, Illumina TruSeq SBS Kit v4-HS reagents, paired-end approach (2 × 150 bp) with 40–55 million reads per sample].

**Genome and transcriptome alignment.** ONT reads were aligned to the human genome (Ensembl primary assembly GRCh38) and transcriptome (combined cDNA and ncRNA reference fasta files from Ensembl GRCh38.90) using minimap2 v2.12[28]. The genome alignments were performed with the arguments −ax splice −N 10, to allow spliced alignments and up to 10 secondary alignments per read. Alignment files from minimap2 were converted to bam format, sorted and indexed using samtools v1.6[29]. The Bioconductor package GenomicAlignments (v1.32.0)[30] was used to extract junctions from the alignments. For each observed junction, we calculated the distance (the absolute difference between the start positions plus the absolute difference between the end positions) to the closest annotated junction. For the transcriptome alignment, we used the arguments −ax map-ont −N 100 to allow more secondary alignments, given the high similarity among transcript isoforms. The minimap2 −p argument, representing the minimal ratio of the secondary to primary alignment score that is allowed in order to report the secondary mapping, has a default value of 0.8. For transcriptome alignment, we investigated the effect of increasing this value in order to restrict the number of reported suboptimal secondary alignments. To evaluate the alignments, we recorded the alignment rates, defined as the fraction of reads with a reported primary alignment, as well as the aligned fraction of each read, which we defined as the sum of the number of M and I characters in the CIGAR string, divided by the full length of the read. In addition to the average base quality across all bases in a read, as reported in the FASTQ files, we estimated the base-level accuracy from the primary genome alignments[18,31], as $(nbrM + nbrI + nbrD − NM)/(nbrM + nbrI + nbrD)$, where nbrM, nbrI, and nbrD are the number of M, I, and D characters in the CIGAR string, respectively, and NM is the edit distance as reported by minimap2.

For some reads, minimap2 also reported supplementary alignments. For each supplementary genome alignment, we compared the alignment position to that of the corresponding primary alignment, and recorded whether these were on the same or different chromosome and/or strand, and whether the primary and supplementary alignments overlapped each other. For each library, we further extracted 100,000 reads randomly, and used the findPalindromes() function in the Biostrings Bioconductor package (v2.48.0) to search for perfect-match palindromes, that is, the presence of a sequence (at least 10 bases long) and its perfect reverse complement in any position in the same read. For each read, we recorded the length of the longest such sequence (for reads where no sequence of at least 10 bases could be found, we assigned a value of 0).

Finally, we generated reduced FASTQ files by retaining only reads with a primary alignment to the genome, and for each such read, removing all bases that were (soft-)clipped in the primary alignment. The resulting bam files were converted to FASTQ format using bedtools bamtofastq v2.27.0[32], and the reads

were subsequently shuffled using bbmap v38.02 (https://sourceforge.net/projects/bbmap/). RSeQC v2.6.5[33] was used to examine the coverage profile along gene bodies for each library, based on the GENCODE basic v24 bed file downloaded from https://sourceforge.net/projects/rseqc/files/BED/Human_Homo_sapiens/ on October 23, 2018.

To investigate to what extent individual ONT reads could be expected to represent full-length transcripts, we selected the best target transcript for each read, starting from the set of all primary and secondary transcriptome alignments obtained with minimap2, with −p set to 0.99. For each read, we kept all alignments for which the number of aligned nucleotides was at least 90% of the maximal such number across all alignments for the read, and among these, we selected the one with the largest transcript coverage degree (number of M and D characters in the CIGAR string of the alignment, divided by the annotated transcript length). While this alignment does not necessarily represent the true origin of the read, the procedure gives an upper bound of the degree of transcript coverage achieved by individual reads.

**Gene and transcript abundance estimation.** Four different computational methods were used to estimate transcript and gene abundances for the ONT libraries. First, we applied Salmon v0.11.0[34] in quasi-mapping mode, with an index generated from the combined Ensembl cDNA and ncRNA reference fasta files and using the default $k$ value of 31 (denoted salmon31 below). For comparability across pipelines, we retained any duplicate transcripts in the index generation. The mean and maximal fragment lengths were set to 600 and 230,000, respectively, and the flag --dumpEq was set to retain equivalence class information. Salmon was also run in quasi-mapping mode on the modified FASTQ files, containing only the aligned part of the primary alignments as described above. Second, we applied Salmon in alignment-based mode to the output bam files from the minimap2 transcriptome alignment, using the flag --noErrorModel to disable the default short-read error model of Salmon in the quantification (denoted salmonminimap2). Third, we applied the bam_count_reads.py script from the Wub package (https://github.com/nanoporetech/wub) to the output files from the transcriptome alignment, setting the minimal mapping quality (−a argument) to 5 (denoted wubminimap2). Finally, we applied featureCounts (from subread v1.6.0)[35,36] to the primary genome alignments, requiring a minimum overlap of 10 bases and using the −L argument to enable the long-read mode (denoted fCminimap2primary). While the Salmon variants and Wub provided transcript-level abundance estimates, which were also aggregated to the gene level, featureCounts provided only gene-level counts and was therefore not considered for transcript quantification.

**De novo transcript identification.** In addition to the reference-based quantification described above, we also performed reference-free, de novo transcript identification using FLAIR (obtained from https://github.com/BrooksLabUCSC/flair on April 18, 2019), applied to the combined primary genome alignments from all libraries in each ONT data set. The minimap2 bam files were converted to bed format using the bam2bed12.py script provided with FLAIR, and identified junctions were subsequently corrected by comparison with either the reference annotation, or both the reference annotation and junctions covered by at least five uniquely mapped reads in the Illumina libraries (for the HAP samples), using the default window size of 10. Next, the corrected reads were collapsed using FLAIR, requiring that the 5′ end of the read falls close to a promoter and retaining only transcripts represented by at least 3 reads. The promoter bed file was obtained by combining active, weak, and poised promoters identified in nine cell lines by the ENCODE consortium (obtained from https://genome.ucsc.edu/cgi-bin/hgFileUi?db = hg19&g = wgEncodeBroadHmm and lifted over to hg38 coordinates using the UCSC Genome Browser liftOver tool). The identified transcripts from each data set were compared with the set of annotated transcripts using gffcompare (https://ccb.jhu.edu/software/stringtie/gffcompare.shtml) and SQANTI v1.2[19], whereby each FLAIR transcript was assigned a class code (gffcompare) or a structural category (SQANTI), detailing the way in which it is related to the most similar reference transcript. For the HAP libraries, SQANTI was provided with the junctions observed in the Illumina libraries.

**PolyA tail length estimation.** We used the nanopolish-polya pipeline[18] (cloned from https://github.com/nanoporetech/pipeline-polya-ng on April 19, 2019) as well as the tailfindr R package (v0.1.0)[22] to estimate polyA tail lengths of the basecalled reads from one of the ONT-RNA001-HAP libraries. The assignment of reads to transcripts by Nanopolish was used to group reads by the transcript biotype annotated in the Ensembl catalog.

**Processing of Illumina libraries.** Sequencing adapters were removed from the Illumina libraries with TrimGalore! v0.4.4 (http://www.bioinformatics.babraham.ac.uk/projects/trim_galore/, using cutadapt v1.13[37]), with quality and length cut-offs both set to 20, and reads were aligned to the Ensembl GRCh38.90 primary genome assembly using STAR v2.5.1b[38]. Abundances of annotated transcripts were estimated using two different methods: first, with StringTie v1.3.3b[39] using reads aligned with HISAT2 v2.1.0[40] (with the --dta flag set and using a known splice site file), and second, with Salmon in quasi-mapping mode, using the same index as for the ONT libraries, and including adjustments for GC content and sequence

bias. Abundances were read into R using tximport (v1.8.0)[41]. In addition, we used StringTie to assemble new transcripts (without the −e flag, provided with the reference gtf file) for comparison with the transcripts identified by FLAIR from the ONT libraries. For this analysis, we merged the HISAT2 bam files from all four Illumina samples to use as the input for StringTie. We used the default coverage cutoff of 2.5 to determine which assembled transcripts to retain in the output file. Assembled transcripts were then characterized with gffcompare and SQANTI, as for the transcripts identified from the ONT data.

**Public data**. In addition to the ONT and Illumina data generated in-house, we processed the SIRV E0 (SRA accession number SRR6058584) and ERCC Mix1 (SRA accession number SRR6058582) ONT dRNA libraries from Garalde et al.[15]. The reads were aligned to the respective transcriptomes using minimap2 with the same settings as above. The SIRV data set was also aligned to the corresponding genome using minimap2 with the settings described above, and additionally setting −−splice-flank = no to accommodate the non-canonical splice sites present in this data. We also downloaded the native RNA PASS reads from the NA12878 cell line sequenced in Workman et al.[18] from https://github.com/nanopore-wgs-consortium/NA12878/blob/master/RNA.md, and aligned them to the genome and transcriptome using the same parameters as for our libraries.

**Reporting summary**. Further information on research design is available in the Nature Research Reporting Summary linked to this article.

## Data availability

The raw sequence files have been uploaded to ArrayExpress under accession numbers E-MTAB-7757 (Illumina) and E-MTAB-7778 (ONT). In addition, we processed the SIRV E0 (SRA accession number SRR6058584) and ERCC Mix1 (SRA accession number SRR6058582) ONT dRNA libraries from Garalde et al.[15], and the native RNA PASS reads from the NA12878 cell line sequenced in Workman et al.[18] (downloaded from https://github.com/nanopore-wgs-consortium/NA12878/blob/master/RNA.md). The source data underlying Figs. 2a–c, 4a–e, 5a–b, 6a–b and Supplementary Figs 1a, 4a-c, 7a-b, 14a-b, 15, 17a-b, 18a-b, 19a-b, 20a-b, 21a-b, 22a-b, 23a-b, 24b, 25 are provided as a Source Data file. The code used to perform the analyses in the paper is available on GitHub: https://github.com/csoneson/NativeRNAseqComplexTranscriptome. Any additional relevant data are available from the authors upon reasonable request.

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

## Acknowledgements

The authors would like to thank Botond Sipos, Michael Stadler and Giovanni d'Ario for constructive discussions and feedback on the paper. Research in the S.H. laboratory was funded by responsive mode project grants BB/N000749/1 and BB/R006431/1 from the Biotechnology and Biosciences Research Council, UK. C.S. was supported by a Pilot

Project grant from the University Research Priority Program Evolution in Action of the University of Zurich. M.D.R. acknowledges support from the University Research Priority Program Evolution in Action at the University of Zurich and the Swiss National Science Foundation (310030_175841).

## Author contributions

C.S.: Conceptualization, data curation, formal analysis, funding acquisition, methodology, software, visualization, writing—original draft, writing—review and editing. Y.Y.: Formal analysis. A.B.-N.: Methodology, writing—review and editing. A.P.: Methodology. M.D.R.: Conceptualization, Data curation, formal analysis, funding acquisition, methodology, writing—review and editing. S.H.: Conceptualization, funding acquisition, investigation, methodology, writing—original draft, writing—review and editing.

## Additional information

**Competing interests:** The authors declare no competing interests.

