## [Peer Review File · Nature Communications]

Reviewers' Comments:

Reviewer #1:

Remarks to the Author:

It is now possible to perform native RNA-seq at a large scale using Nanopore technology. This technology is now finding wide use in the genomics community with a wide range of applications from de novo genome assembly to transcriptomics and in the field pathogen detection. The technology is offering an open development platform for signal processing as well as a opportunities to develop a variety of tailored bioinformatics tools, which has inspired the scientific community. The possibility to sequence RNA molecules directly, without the need to synthesize complementary bases represents a major significant advance in genomics.

Within this context, the authors performed a detailed evaluation of RNA-seq sequencing protocols utilizing RNA from two different human cell lines and three different nanopore kits, two based on cDNA synthesis and one native RNA sequencing kit. PCR amplification was not included in order to measure relative performance directly, as native RNA-seq is amplification free. The work is done with great care and extensive data analysis outputs are provided, comparing the relative performance of the three types of kits, including also a comparison to standard Illumina short read based RNA-seq.

The manuscript provides detailed information on alignment to genome and transcriptome, metrics on coverage of full-length transcripts, detection of transcripts and abundance estimation. In terms of transcript detection and accurate identification, the authors perform extensive in-depth analysis to compare a reference-based method to reference free identification utilizing reference information only in part (for example reads with a 5' close to established promoter sequences) to generate high-confidence consensus transcripts.

All these metrics and the compilation of tools utilized represent a valuable addition to the nanopore sequencing community and can be used as a guide for the analysis of nanopore derived RNA-seq data. All tools used are summarized and accessible through the web, a very welcome feature of the manuscript.

The main conclusions can be summarized as follows:

1. Transcript isoform identification with nanopore reads remains a challenge, particularly because the mapping scores between primary and secondary alignments are very close.
2. Native RNA-seq reads in particular do not represent full length transcripts. This is mainly due to difficulties in obtaining sequence from the first 10-15 bases at the 5' end of the RNA, as well as other factors that contribute to high error rates.
3. One important point is that the sequencing depths achieved with the native RNA-seq libraries was not enough to saturate transcript detection, a point the authors highlight in the discussion. In general, the libraries prepared by a template switching (Smart) approach led to the highest yields in terms of read numbers/flowcell.
4. The performance of the non- native RNA-seq based approaches was similar to already published benchmark studies.

The conclusions of the paper are not particularly favourable in terms of the usefulness of native RNA-seq using nanopores considering the requirements of big amounts of total RNA (in order to isolate 100ng of polyA RNA); and the full potential of the method is difficult to achieve due to lower read yields reported in here, per library and flowcell compared to template switching based cDNA sequencing.

The authors do not explore one important reason for performing such an experiment, which is the detection of RNA modifications as well as the measurement of polyA tail length, a critical regulatory feature in early development.

This type of analysis needs to be added.

As such the paper remains largely prescriptive, albeit with very thorough analysis of the data in terms of alignment and transcript identification, and will be of interest to a more specialized audience.

Minor points:

1. The read numbers obtained in the different runs using the non smart approaches vary considerably. Please indicate whether each library was sequenced once and therefore each dataset is derived from individual library preps.
2. The figures are too detailed for main text figures. In most cases one should be able to summarize performance metrics from all reads per library type, for example in figure 3, figure 5A etc, instead of having a large number of separate figures for each run. This level of detail is more suitable for supplementary data.

Reviewer #2:

Remarks to the Author:

In their manuscript "A comprehensive examination of Nanopore native RNA sequencing for characterization of complex transcriptomes" Sonesson, et al. benchmark the performance of 3 different library preparation protocols for transcriptome analysis. This benchmarking includes the alignment characteristics of sequencing reads, their ability to cover entire transcripts, their ability to identify annotated genes and transcripts as expressed, as well as identify new transcripts in a reference-free manner.

The main findings we took away from this manuscript are that 1.) cDNA and native reads are limited in their ability to cover full-length transcripts completely, especially if the transcripts are long (>2kb). 2.) The DCS108 cDNA protocol seems to be burdened with a large fraction of potential 1D2 artifacts. 3.) Native RNA sequencing may not have the throughput for comprehensive transcriptome analysis.

This manuscript will no doubt be a useful resource to the growing field of long-read transcriptomics by identifying remaining challenges when using both cDNA and native RNA based ONT sequencing approaches.

There is in fact an urgent need for independent validation of ONT sequencing technology and this manuscript is a good companion to another manuscript by Workman et al. currently on biorxiv (and mentioned by the authors) which analyzed a different cell line using native RNA and the PCR-based PCS108 cDNA protocol.

There are however several major and some minor issues with this manuscript both in the analysis and presentation of the data that hamper its usefulness which we will outline below.

Major issues:

1) Analyzing individual technical replicates separately is not informative and makes figures very hard to interpret. Unless replicates show a meaningful qualitative difference (which they don't seem to based on the data and figures shown) their reads should be summarized (Table instead of Figure 2AB) and combined. This would massively simplify analysis and figures (5AB, and 6ABCD) and allow for higher read depth for analysis. To determine the effects of varying read depth, subsampling of this combined data should be performed. Some of this was done in Fig. S11 but this should be the major mode of analysis. This is especially important because at ~2.5 million RNA reads per cell line, even the combined datasets presented by the authors are fairly shallow.

2) The manuscript spends quite a lot of time discussing a modified 1D version of the NSK007 "2D" kit. In its current iteration this modified protocol is of very limited interest to the field,

because the NSK007 kit is no longer offered by ONT. We sympathize with the authors because ONT is known to discontinue sequencing kits fairly often and this can be highly frustrating to academic researchers. However, because the transcriptome sequencing field will be unable to replicate this work and because the titular focus of this manuscript is native RNA sequencing, we strongly recommend this work be omitted or moved mostly into the supplement.

3) Inclusion of direct RNA data on the HEK293 cell line data is also of questionable usefulness. The other protocols were only applied to the HAP1 cell line making direct comparison challenging. There also seems to be very little difference in read characteristics between HEK293 and HAP1 data. In order to streamline the manuscripts, I would again recommend to move this data into the supplement.

4) Figure 2C alludes to the potential 1D2 artifacts in cDNA data. This has large implications for data analysis and should be explored further. Do reads with supplemental alignments consistently contain palindromes, i.e. template and reverse complement of the same read? The LAST aligner could be used to explore this anecdotally but a systematic approach to identify these palindromes would be preferable.

5) Inclusion of Workman, et al. data in Figure 4. Garalde, et al. data shown in Figure 4 seems to indicate that native RNA sequencing performed by the author may be not quite representative of other labs work although the synthetic RNA analyzed in the Garalde paper was short. The public availability of ~10 million native RNA sequencing reads covering human cell line RNA should be taken advantage of to see whether the pattern of incomplete coverage of long transcripts holds up between labs.

Minor issues:

1) Average base quality of cDNA runs seems inflated/"off" in all figures. How was quality determined? If this basequality was determined based on the fastq quality string, this should be replaced by accuracy ($\text{Matches}/(\text{Matches}+\text{Mismatches}+\text{Indels})$) based on the edit-distance of the read to its primary genomic alignment. This can be done fairly easily using the edit-distance "NM:i:" tag in minimap2 sam files. This should also clear up the cDNA vs RNA quality comparison mentioned in line 117 which states that cDNA reads are more accurate than RNA reads when the Workman et al. paper shows the opposite.

2) By omitting NSK007, RNA-HEK and combining replicates, figure 3 could be reduced to 2 scatter plots without losing any meaningful information.

3) Across the manuscript, transcript length is shown on logarithmic scale although any transcripts or reads beyond 15kb appear to be clear outliers. I recommend limiting the scale to 0-15kb and making them linear. Figure 4D in particular would also benefit greatly from separating the individual lines into their own violin plots.

4) Den-novo transcript identification. Categorizing transcripts identified de-novo should use Sqanti instead of cuffcompare. The "novel, in catalog" category could reveal transcripts that don't match an annotated transcript but uses only known splice junctions. Further, FLAIR recommends the correction of splice junction in RNA reads with Illumina data which is available to the authors. However, it appears that this step was not performed. Performance with this correction would be of interest to scientist considering native RNA sequencing.

5) Line 215 "begin" should probably read "being"

6) N50 values (Line 123) are mostly meaningless in transcriptome sequencing. Median, maybe quartiles, is really all you need to know.

7) The fact that 500,000 reads are not enough for comprehensive transcriptome analysis does not come as a surprise and neither should it be treated as such. For expressed gene identification, a 150bp Illumina read that uniquely aligns is worth just as much as a 900bp native RNA read and nobody would consider doing a 500,000 read Illumina RNA-seq experiment.

We greatly appreciate the reviewers' encouraging comments, and we would like to thank them for their thoughtful and constructive feedback. In the revised manuscript, we have addressed the reviewers' comments and incorporated suggested changes, as described in the point-by-point response below (our responses are provided in *italic*). Major changes are also indicated with blue text in the main manuscript.

Reviewer #1 (Remarks to the Author):

It is now possible to perform native RNA-seq at a large scale using Nanopore technology. This technology is now finding wide use in the genomics community with a wide range of applications from de novo genome assembly to transcriptomics and in the field pathogen detection. The technology is offering an open development platform for signal processing as well as a opportunities to develop a variety of tailored bioinformatics tools, which has inspired the scientific community. The possibility to sequence RNA molecules directly, without the need to synthesize complementary bases represents a major significant advance in genomics.

Within this context, the authors performed a detailed evaluation of RNA-seq sequencing protocols utilizing RNA from two different human cell lines and three different nanopore kits, two based on cDNA synthesis and one native RNA sequencing kit. PCR amplification was not included in order to measure relative performance directly, as native RNA-seq is amplification free. The work is done with great care and extensive data analysis outputs are provided, comparing the relative performance of the three types of kits, including also a comparison to standard Illumina short read based RNA-seq.

The manuscript provides detailed information on alignment to genome and transcriptome, metrics on coverage of full-length transcripts, detection of transcripts and abundance estimation. In terms of transcript detection and accurate identification, the authors perform extensive in-depth analysis to compare a reference-based method to reference free identification utilizing reference information only in part (for example reads with a 5' close to established promoter sequences) to generate high-confidence consensus transcripts.

All these metrics and the compilation of tools utilized represent a valuable addition to the nanopore sequencing community and can be used as a guide for the analysis of nanopore derived RNA-seq data. All tools used are summarized and accessible through the web, a very welcome feature of the manuscript.

The main conclusions can be summarized as follows:

1. Transcript isoform identification with nanopore reads remains a challenge, particularly because the mapping scores between primary and secondary alignments are very close.
2. Native RNA-seq reads in particular do not represent full length transcripts. This is mainly due to difficulties in obtaining sequence from the first 10-15 bases at the 5' end of the RNA, as well as other factors that contribute to high error rates.
3. One important point is that the sequencing depths achieved with the native RNA-seq libraries was not enough to saturate transcript detection, a point the authors highlight in the discussion. In general, the libraries prepared by a template switching (Smart) approach led to the highest yields in terms of read numbers/flowcell.
4. The performance of the non- native RNA-seq based approaches was similar to already published benchmark studies.

The conclusions of the paper are not particularly favourable in terms of the usefulness of native RNA-seq using nanopores considering the requirements of big amounts of total RNA (in order to isolate 100ng of polyA RNA); and the full potential of the method is difficult to achieve due to lower read yields reported in here, per library and flowcell compared to template switching based cDNA sequencing.

The authors do not explore one important reason for performing such an experiment, which is the detection of RNA modifications as well as the measurement of polyA tail length, a critical regulatory feature in early development.

This type of analysis needs to be added.

As the reviewer notes, Nanopore native RNA sequencing has promising applications beyond identification and quantification of transcripts and genes, e.g., for detection of RNA modifications and estimation of polyA tail lengths. In this manuscript, our main focus is on the transcript identification and quantification aspects, a choice that we made in order to enable a comprehensive investigation, and since this is likely to be of significant interest to a broad readership. Thus, while the additional aspects mentioned by the reviewer are definitely exciting, performing an equally comprehensive investigation of them, in our opinion, falls outside the scope of the current manuscript. In the revised manuscript, however, we do include an analysis of estimated polyA tail lengths in one of the native RNA libraries. We estimate polyA tail length using two different methods (Nanopolish and tailfindr). Our results corroborate those from previous studies, for example, showing that reads mapping to mitochondrial genes have generally shorter polyA tails. Overall, our observations suggest that polyA tail length estimation using Nanopore native RNA sequencing might already be considered an optimal method among the currently available relevant techniques (lines 441-462, 508-513).

Regarding RNA modification detection, ONT offer a suitable analysis pipeline, 'Tombo', available via GitHub (https://nanoporetech.github.io/tombo/modified_base_detection.html) so far only for the detection of the RNA m5C modification type, which is in fact very sparse in mammalian messenger RNA (Legrand et al. 2017). Preliminary investigations in the laboratory of one of us (Hussain) indicate that, in its current form, the Tombo RNA m5C base caller likely yields a very significant proportion of false positive detections when applied on a transcriptome-wide scale. Furthermore, it is now consensus opinion within the epitranscriptomics field that modification sites detected by sequencing-based studies require additional validation via orthogonal biochemical methods (Helm and Motorin 2017; Grozhik and Jaffrey 2018). We believe that such a line of investigation is well beyond the scope of this particular current study, although we now provide some discussion of these exciting prospects of Nanopore sequencing within the manuscript (lines 514-521).

Legrand et al. 2017. *Statistically robust methylation calling for whole-transcriptome bisulfite sequencing reveals distinct methylation patterns for mouse RNAs. Genome Res., 27(9):1589-1596. doi: 10.1101/gr.210666.116*

Helm and Motorin 2017. *Detecting RNA modifications in the epitranscriptome: predict and validate. Nat Rev Genet., 18(5):275-291. doi: 10.1038/nrg.2016.169*

Grozhik and Jaffrey 2018. *Distinguishing RNA modifications from noise in epitranscriptome maps. Nat Chem Biol., 14(3):215-225. doi: 10.1038/nchembio.2546*

As such the paper remains largely prescriptive, albeit with very thorough analysis of the data in terms of alignment and transcript identification, and will be of interest to a more specialized audience.

Minor points:

1. The read numbers obtained in the different runs using the non smart approaches vary considerably. Please indicate whether each library was sequenced once and therefore each dataset is derived from individual library preps.

Across the study, each individual replicate was derived from individual library preps; each library was sequenced once. We have adjusted the Methods to clarify this (lines 550-555).

2. The figures are too detailed for main text figures. In most cases one should be able to summarize performance metrics from all reads per library type, for example in figure 3, figure 5A etc, instead of having a large number of separate figures for each run. This level of detail is more suitable for supplementary data.

In the revised manuscript, we have simplified all the main figures to show only results summarized by library type. In addition, data sets less directly relevant to specific conclusions have been excluded from the corresponding main figures (Figures 2E and 6B-C) for simplicity, and some of the previous main figures have been moved to the supplement, with the aim of also illustrating the variability between the individual libraries.

Reviewer #2 (Remarks to the Author):

In their manuscript "A comprehensive examination of Nanopore native RNA sequencing for characterization of complex transcriptomes" Sonesson, et al. benchmark the performance of 3 different library preparation protocols for transcriptome analysis. This benchmarking includes the alignment characteristics of sequencing reads, their ability to cover entire transcripts, their ability to identify annotated genes and transcripts as expressed, as well as identify new transcripts in a reference-free manner.

The main findings we took away from this manuscript are that 1.) cDNA and native reads are limited in their ability to cover full-length transcripts completely, especially if the transcripts are long (>2kb). 2.) The DCS108 cDNA protocol seems to be burdened with a large fraction of potential 1D2 artifacts. 3.) Native RNA sequencing may not have the throughput for comprehensive transcriptome analysis.

This manuscript will no doubt be a useful resource to the growing field of long-read transcriptomics by identifying remaining challenges when using both cDNA and native RNA based ONT sequencing approaches.

There is in fact an urgent need for independent validation of ONT sequencing technology and this manuscript is a good companion to another manuscript by Workman et al. currently on bioRxiv (and mentioned by the authors) which analyzed a different cell line using native RNA and the PCR-based PCS108 cDNA protocol.

There are however several major and some minor issues with this manuscript both in the analysis and presentation of the data that hamper its usefulness which we will outline below.

Major issues:

1) Analyzing individual technical replicates separately is not informative and makes figures very hard to interpret. Unless replicates show a meaningful qualitative difference (which they don't seem to based on the data and figures shown) their reads should be summarized (Table instead of Figure 2AB) and combined. This would massively simplify analysis and figures (5AB, and 6ABCD) and allow for higher read depth for analysis. To determine the effects of varying read depth, subsampling of this combined data should be performed. Some of this was done in Fig. S11 but this should be the major mode of analysis. This is especially important because at ~2.5 million RNA reads per cell line, even the combined datasets presented by the authors are fairly shallow.

In the revised manuscript, we have aggregated the reads across all libraries of a given type in all relevant main figures. Thus, all main figures are now based on a larger read depth, and some of the previous main figures have been moved to the supplement to illustrate the variability among replicates. Most of the results in the manuscript are not directly influenced by the read depth, since they are largely concerned with the fraction of the reads that show a certain property. Thus, subsampling reads randomly will not change those conclusions (this can also be seen by comparing the new, aggregated main figures to the corresponding supplementary figures showing the individual replicates). The exception is the investigation of the number of detected transcripts and genes, for which we already perform a subsampling analysis, as noted by the reviewer.

2) The manuscript spends quite a lot of time discussing a modified 1D version of the NSK007 "2D" kit. In its current iteration this modified protocol is of very limited interest to the field, because the NSK007 kit is no longer offered by

ONT. We sympathize with the authors because ONT is known to discontinue sequencing kits fairly often and this can be highly frustrating to academic researchers. However, because the transcriptome sequencing field will be unable to replicate this work and because the titular focus of this manuscript is native RNA sequencing, we strongly recommend this work be omitted or moved mostly into the supplement.

We agree with the reviewer that the observations pertaining to this protocol may be of lower direct practical utility. However, from our point of view it illustrates some important points, not least in the comparison to the ONT-DCS108 protocol. In particular, we can show that the template switching in the ONT-DCS108 protocol indeed leads to a larger fraction of full-length transcripts being captured. For comparative purposes, we have therefore retained it in the relevant main figures. However, to simplify the narrative, we have excluded the NSK007 data from Figures 2E and 6B-C.

3) Inclusion of direct RNA data on the HEK293 cell line data is also of questionable usefulness. The other protocols were only applied to the HAP1 cell line making direct comparison challenging. There also seems to be very little difference in read characteristics between HEK293 and HAP1 data. In order to streamline the manuscripts, I would again recommend to move this data into the supplement.

While we did not perform cDNA ONT sequencing or Illumina sequencing for the HEK293 cell line, in our opinion it still serves a valuable purpose in the manuscript, for example by allowing us to see that the transcripts identified by FLAIR were sometimes more similar between the two native RNA data sets than between the data sets from the same cell line, but acquired with different library preparation kits, suggesting that the library preparation choice can have important effects on the results. In addition, we can show that the characteristics that we observe for the native RNA sequencing reads are not specific to the HAP1 cell line. For simplicity, we have excluded the HEK293 libraries from some of the main figures (2E, 6B-C).

4) Figure 2C alludes to the potential 1D2 artifacts in cDNA data. This has large implications for data analysis and should be explored further. Do reads with supplemental alignments consistently contain palindromes, i.e. template and reverse complement of the same read? The LAST aligner could be used to explore this anecdotally but a systematic approach to identify these palindromes would be preferable.

We have now performed a further analysis of the relevant supplementary alignments by measuring the extent of overlap between the primary and supplementary alignments (which occur on opposite strands). In our view, this provides strong evidence for the reverse complement forming nature of these reads (Figure 2D). Due to the lower accuracy of Nanopore reads, an analysis of palindrome content is non-trivial; however, we did also perform some investigation of palindrome presence (i.e., the presence of any (>10bp) sequence as well as its perfect reverse complement in the same read), and the length of the longest such palindrome in the reads. We note that the ONT-DCS108-HAP reads contain considerably longer palindromes than the ONT-RNA001-HAP reads, and further that within the ONT-DCS108-HAP libraries, reads with supplementary alignments tend to contain longer palindromes than reads without supplementary alignments (Supplementary Figure 9).

5) Inclusion of Workman, et al. data in Figure 4. Garalde, et al. data shown in Figure 4 seems to indicate that native RNA sequencing performed by the author may be not quite representative of other labs work although the synthetic RNA analyzed in the Garalde paper was short. The public availability of ~10 million native RNA sequencing reads covering human cell line RNA should be taken advantage of to see whether the pattern of incomplete coverage of long transcripts holds up between labs.

Thank you for this suggestion. In the revised manuscript, we have included the native RNA reads from Workman et al in this figure (now Figure 3). Overall, we see a similar pattern of incomplete coverage of long transcripts also in this dataset.

Minor issues:

1) Average base quality of cDNA runs seems inflated/"off" in all figures. How was quality determined? If this basequality was determined based on the fastq quality string, this should be replaced by accuracy ($\text{Matches}/(\text{Matches}+\text{Mismatches_Indels})$) based on the edit-distance of the read to its primary genomic alignment. This can be done fairly easily using the edit-distance "NM:i:" tag in minimap2 sam files. This should also clear up the cDNA vs RNA quality comparison mentioned in line 117 which states that cDNA reads are more accurate than RNA reads when the Workman et al. paper shows the opposite.

The average base qualities were obtained from the fastq quality strings. As such, they can be calculated for all reads and bases (regardless of whether they align or not). In the revised manuscript, we have added an estimate of the accuracy (calculated from the CIGAR strings and NM tags of the primary genome alignments as $(\text{nbrM} - \text{NM} + \text{nbrI} + \text{nbrD})/(\text{nbrM} + \text{nbrI} + \text{nbrD})$) to Supplementary Figure 5E. We have also added Supplementary Figure 3, showing a strong agreement between the average base quality and the estimated accuracy, for most samples. As expected, the largest deviation is seen for the DCS108 libraries, where many of the bases are soft clipped in the primary alignments and thus the two accuracy measures are based on different sets of bases. To summarise, while the Workman et al. study found 86% and 85% mean identities for aligned RNA and cDNA-PCS108 reads respectively, our corresponding calculated values were NSK007: 85%, DCS108: 80%, RNA001-HAP: 83%, RNA001-HEK: 83%. The precise reasons for the subtle differences across the two studies are unclear to us.

2) By omitting NSK007, RNA-HEK and combining replicates, figure 3 could be reduced to 2 scatter plots without losing any meaningful information.

Thank you for this suggestion. We have combined replicates and only included the ONT-RNA001-HAP and ONT-DCS108-HAP data sets in this figure, which is now Figure 2E.

3) Across the manuscript, transcript length is shown on logarithmic scale although any transcripts or reads beyond 15kb appear to be clear outliers. I recommend limiting the scale to 0-15kb and making them linear. Figure 4D in particular would also benefit greatly from separating the individual lines into their own violin plots.

As suggested, we have modified Figure 4D (now Figure 3E) to instead show violin plots. However, after careful consideration, we find that the logarithmic scale allows easier interpretation of the graphs, and at the same time avoids an arbitrary length cutoff for the reads and/or transcripts.

4) Den-novo transcript identification. Categorizing transcripts identified de-novo should use Sqanti instead of cuffcompare. The "novel, in catalog" category could reveal transcripts that don't match an annotated transcript but uses only known splice junctions. Further, FLAIR recommends the correction of splice junction in RNA reads with Illumina data which is available to the authors. However, it appears that this step was not performed. Performance with this correction would be of interest to scientist considering native RNA sequencing.

In the revised manuscript, we now use SQANTI to assign transcripts to a structural category in Figures 6-7. The gffcompare assignments, as well as a comparison between the two classification schemes, have been included in the Supplementary Material for completeness (Supplementary Figures 23-25). In addition, we updated our analysis to use a newer version of FLAIR (from April 19), and ran it both with and without including the splice junctions observed in the Illumina libraries.

5) Line 215 "begin" should probably read "being"

Thank you for spotting this typo, it has been corrected.

6) N50 values (Line 123) are mostly meaningless in transcriptome sequencing. Median, maybe quartiles, is really all you need to know.

We have removed the N50 values.

7) The fact that 500,000 reads are not enough for comprehensive transcriptome analysis does not come as a surprise and neither should it be treated as such. For expressed gene identification, a 150bp Illumina read that uniquely aligns is worth just as much as a 900bp native RNA read and nobody would consider doing a 500,000 read Illumina RNA-seq experiment.

We agree to a large extent, and have adjusted the relevant text in the Discussion to better convey our opinion.

Reviewers' Comments:

Reviewer #1:

Remarks to the Author:

The authors attempted to address the reviewer's points. In general, the manuscript is now clearer and the figures concentrate on presenting the key data points.

The authors were not able to add work on RNA modification analysis, nor on editing and for example whether editing (A to I) can be detected in direct RNA sequencing compared to cDNA sequencing.

This time an attempt to perform measurements of polyA tail length has been made, at least as a proof of principle. Results are now presented at transcript class level, showing a polyA length difference between mitochondrial and other species.

Point: reference 21 includes detailed reports on mitochondrial transcript polyA tail lengths and HEK-293 cells have been analyzed. However, in figure 8C it is very difficult to see where the mitochondrial median lies, a mark indicating 40bp would be useful.

The minor points were addressed.

Reviewer #2:

Remarks to the Author:

I believe that the changes by the authors, in particular the simplification of the figures by relegating the analysis of technical replicates to the supplement, strengthened the manuscript considerably.

The manuscript now does a good job outlining the strengths and weaknesses of direct RNA sequencing which is information sorely needed by the long-read sequencing field.

I also really appreciate the inclusion of the NA12878 data to connect the different direct RNA sequencing characterization efforts currently going on in the field.

Inclusion of Sqanti in the analysis now re-iterates the point that many novel transcripts are likely incomplete.

The inclusion of polyA analysis is interesting as well.

I do agree with the authors that modification detection in direct RNA reads is not ready for prime-time just yet.

Finally, while the authors did not address all my requests, I think they sufficiently justified the cases where they did not.